# Plant Resource Use and Pattern of Usage by the Naturalized Orchid Bee (*Euglossa dilemma:* Hymenoptera: Apidae) in Florida

**DOI:** 10.3390/insects14120909

**Published:** 2023-11-27

**Authors:** Robert W. Pemberton

**Affiliations:** Independent Researcher, 2275 1st Ave NE, Atlanta, GA 30317, USA; rpemberton5@gmail.com

**Keywords:** buzz pollination, Euglossine bee, floral rewards, fragrance chemicals, pollinators

## Abstract

**Simple Summary:**

*Euglossa dilemma*’s naturalization in Florida has provided a unique opportunity to study the ecology and biology of an orchid bee distant from its native region. Prior studies have demonstrated its uniqueness as a highly specialized pollinator of invasive plants in Florida. This study documents its use of 259 plant taxa, 237 species and 22 horticultural forms in 156 genera, and 56 families that it uses for pollen, nectar, resins for its nest construction, and fragrance chemicals needed for its courtship. The presence of the bee in a geographic area distant from its native region has resulted in its use of many novel plants, but, like other studied naturalized bees, most of the plants it uses are native, or closely related to native plants in its native region. Because of the bee’s very long tongue, ability to vibrate pollen out of specialized flowers, and facility in collecting fragrance and resin rewards from specialized flowers, it uses a higher proportion of these flowers than flowers with easily collected rewards. The ability of *E. dilemma* to use a large assemblage of plants, including those in most human habitats and natural communities, has facilitated its spread and colonization. Given *E. dilemma*’s abundance and wide occurrence in subtropical Florida and its use of numerous plant species with flowers with both specialized and generalized pollination systems, it has become a common, and at times, an important pollinator.

**Abstract:**

The Neotropical orchid bee *Euglossa dilemma* was found to be naturalized in southern Florida in 2003, and, by 2022, it had colonized the southern half of Florida. Observations of the bee’s collection of plant resources, primarily flowers, were made from 2003 through to 2022 to document its plant usage and understand the patterns of its plant usage. The bee utilized 259 plant taxa, 237 species, and 22 horticultural forms, in 156 genera and 56 families in 263 total uses. Of 247 taxa of flowers, 120 were visited primarily for nectar, 46 for both nectar and pollen, 60 for pollen, including 42 buzz-pollinated flowers, 15 for fragrance chemicals for the males, and 5 for resin rewards by females for nesting. Fragrance chemicals were also collected by males from the leaves of 12 plant species. These extensive resource use data allowed the following predictions to be made. (1) The bee’s presence in Florida, distant from its native region of Mexico and Central America and the geographical ranges of other orchid bees, would result the usage of many new taxa of plants. True, half, 74/148 (50%), of the genera and one third, 16/51(31%), of the plant families of the plants with flowers used by the bee were not previously recorded as being utilized by Euglossine bees. (2) Like other naturalized bees, it would use relatively more plants from its native range or congeners of these plants. True, 113/148 (76%) of genera with species bearing collected floral rewards are native or congeners with species native to the bee’s native range. (3) Given the bee’s long tongue, ability to buzz pollen from poricidal anthers, and ability to collect and use specialized rewards, it would disproportionately use plants with protected or highly specialized floral rewards. True, 180/247 (72%) utilized species bear rewards which were protected and unavailable to, or of no interest to, most other flower visitors.

## 1. Introduction

Orchid bees (Apidae: Euglossini) have fascinating and diverse relationships with plants. They are pollen generalists but also specialists when seeking other needed resources. Male orchid bees from the estimated 250 species collect fragrance chemicals from particular orchid species, known as perfume orchids, as well as from many other plant and fungal or decay volatiles, and may pollinate flowers in the process [1,2,3,4,5,6,7,8]. Male bees use these compounds as pheromone analogs in their courtship and female orchid bees mate only if the male bees present a specific mix of chemicals [8,9,10,11]. This relationship involves about 600–700 species of neotropical orchids which are completely dependent on male orchid bees for pollination [12]. Although orchid bees were known to collect volatile chemicals from sources other than orchids [8,13,14], they were widely believed to be dependent on their mutualist orchids [10]. The naturalization of *Euglossa dilemma* Bembé & Eltz in Florida by 2003, where these perfume orchids are absent, allowed the orchid dependence proposition to be tested directly. Along with their ability to obtain essential chemicals from leaves and other sources, mutualism was determined to be facultative for the bees [15], as had already been suggested [6,16]. Perfume orchids have evolved at least three times for male Euglossine bee pollination, taking advantage of the bee’s existing behavior to collect volatile chemicals [8]. Perfume flowers have evolved in the Araceae, Annonaceae, Bromeliaceae, Solanaceae, and a few other families for Euglosssine bee pollination [17,18,19,20]. Female orchid bees collect plant resins to build their nest cells and the plants include the uncommon floral resin rewards from some plants such as *Clusia* species (Clusiaceae) and *Dalechampia* (Euphorbiacae) [4,6,21]. Female orchid bees also collect pollen to provide their brood cells via direct mechanical collection and frequently through buzz pollinating specialized flowers of *Solanum* species (Solanaceae), *Senna* (Fabaceae), and members of other families that have poricidal anthers [22]. Orchid bees of both sexes visit a diversity of flowers to obtain nectar for energy and females mix nectar with pollen to provide their brood cells.

Euglossine bees account for a large fraction of the pollination services in the Neotropical region, pollinating thousands of plants [4,13,23,24,25]. The floral resources used by Euglossine bees have been documented by Zucchi et al. [26], Roubik [27], Dressler [4], Ackerman [28], Roubik and Ackerman [17], Kress and Beach [29], Arriaga and Hernández [30], Ramírez et al. [7], Roubik and Hanson [6], Pemberton and Wheeler [15], Rocha-Filho et al. [31], Silva et al. [32], Villanueva-Gutierrez et al. [33], Ferreira-Caliman et al. [34], and Roubik and Moreno [25]. Three of these, Ramírez et al. [7], Roubik and Hanson [6], and Roubik and Moreno [25], are extensive compilations of the literature and author observations. The primary objective of this study was to determine what species of plants this naturalized orchid bee uses in Florida and what their characteristics are, particularly of the rewards that the bee seeks. The secondary aim was to determine what proportion of the used plants are sympatric with the bee in its native geographic area of Mexico and Central America or are congeneric with plants occurring in this native region.

## 2. Materials and Methods

### 2.1. The Study Species

The tropical American orchid bee *E. dilemma* (as *E. viridissima* Friese 1899) was reported to be naturalized in Broward County, Florida, in 2003 as *E. viridissima* Friese [15,35]. At the time of the bee’s discovery in southern Florida, it had already established a large local population [15]. The bee, which is native to Mexico and Central America, was discovered by Eltz and colleagues [36] to be a new sibling species of *E. viridissima,* subsequently named *E. dilemma*. *Euglossa dilemma* has since spread to occupy Florida from the island Keys in the south to central Florida in the north [37]. This orchid bee has recently been found in the Dominican Republic on Hispaniola [38], probably having been accidentally introduced from Florida.

Since first observing a female *E. dilemma* collecting pollen from a *Begonia* cultivar in Fort Lauderdale, Florida, in 2003, the author has been making observations of the flowers and other plant parts used by this bee. A preliminary list of 46 species in 17 families was published in 2006 [15]. The occurrence of single orchid bee species in Florida, with its distinctive appearance, created a unique opportunity to define the plants used by a single bee species and thereby expand our understanding of plant use by Euglossine bees. In tropical America, where there are many orchid bees of similar coloration and size even in the same habitats, it can be difficult to attribute observations of floral visits to a particular bee species, particularly because these bees fly rapidly and are usually less easily approached compared to honeybees or bumble bees. In addition, the identity of the flowers visited in Florida is usually more easily known than those in more species-rich wild habitats in tropical America. The large number of flowers visited by this bee in Florida also provides an opportunity to know more about its patterns of floral use with regard to functional floral characteristics, ecology, and taxonomy. A comparison of the observations of *E. delimma*’s plant use in Florida to those of Euglossine bees is recorded in the literature, enabling many novel floral hosts to be identified and documented. The floral use data were also used to learn if *E. dilemma*, like other introduced bees, prefers plants from its native region [39,40].

### 2.2. Data Collection

Three methods were utilized to document the plants used by *E. dilemma*. Direct observations were made of the bee’s visitation to flowers and other plant parts in natural areas, botanical gardens, plant nurseries, and residential landscapes. Most of the direct observations were made in southeastern Florida in the Broward, Palm Beach, and Miami Dade Counties where the bee has been abundant for most of the 20-year observation period. These observations were made all year because the bee and many potential floral resources are tropical, so are available throughout the year. To learn about the potential use of less common plants and types of plants by the bee, potted plants were introduced to and observed in a residential garden where the bee was abundant. Exposed plants included buzz pollinated plants such as the *Solanum* and *Senna* species; resin reward flowers, the *Clusia* and *Dalechampia* species; and perfume orchids known to be visited by Euglossine bees. After male bees were observed to collect fragrance chemicals from basil leaves (*Ocimum basilicum* L.), other *Ocimum* species were introduced to the garden. Lastly, photographs of *E. dilemma* visiting flowers in Florida posted on the citizen science websites iNaturalist [41] and BugGuide [42] were examined to supplement the author’s observations. If the plant being visited could be identified and it constituted a floral host not previously observed by the author, it was added to the data set. When there was more that one iNaturalist post of *E. dilemma* visiting flowers and other parts of plants, not seen by the author, only the earliest post was used and acknowledged. The use of iNaturalist, with more than 700 postings of photographs of *E. dilemma* in Florida, proved valuable in helping to determine the bee’s current distribution in Florida [37]. INaturalist photograph-based plant use records by *E. dilemma* in Florida are uniquely trustworthy because *E*. *dilemma* is the only orchid bee in Florida, whereas many other green *Euglossa*, or other bees of its general size and color, are present in the neotropics. The visitation frequency of *E. dilemma* to plants was not recorded.

### 2.3. Flower and Plant Characterization

The floral rewards sought or collected by the bee were noted when observed or apparent. When the reward collected could not be observed, such as when the bee disappeared into gullet flowers, Roubik and Hanson’s 2004 compilation of non-orchid genera, containing species visited by orchid bees and their rewards, was followed. Flower form as it related to how the reward was protected and accessed was categorized. These included gullet and tubular forms and buzz-pollinated types. Flowers without those forms were taxonomically and morphologically diverse. These were categorized as “open” when the rewards were exposed, such the pollen in male *Begonia* flowers, or held in shallow readily accessed structures, such as in the florets of flowers in the Asteraceae. Categorized as “closed” were plants such as papilionaceous legume flowers, since they require manipulation to access the rewards. When distinctive, the gender of the visiting bees was noted. The color of solid-colored flowers was recorded subjectively and tallied as such. The life form of the plants bearing visited flowers was noted and these included trees, shrubs, vines and herbs, including epiphytes. The types of plants with visited flowers were categorized as native, ornamental, naturalized, invasive, weed, human food, or a combination of these. If the plant fit more than one type, such as a native plant that was also an ornamental, it was listed as “native/ornamental”. All plants designated as ornamentals had been introduced from other countries unless described as native/ornamental. Both weeds and invasives had also been introduced from other countries, but invasives are harmful to native communities while weeds are not. Ornamentals that had escaped to become naturalized or invasive were noted. For plant use observations not made by the author, the source is given. With few exceptions, these are from author identifications of plants in the photographs of *E. dilemma* visiting flowers posted on iNaturalist and a few posted on BugGuide. To provide a more complete picture of the plants used in Florida, one complete list is presented, incorporating both the preliminary list published by Pemberton and Wheeler [15] and the observations made since that date. Tropical Flowering Plants [43] was helpful in identifying less common ornamental plants, as were some knowledgeable people who are acknowledged. The classification of plants as native or naturalized was verified by the Florida Plant Atlas [44] and KEW’s Plants of the World Online [45]. Species taxonomy is based on Plants of the World Online. Novel plant families and genera, those with species not previously recorded to be used by Euglossine bees, were identified by comparing the families and genera of plants used in this study with those in the aforementioned literature. Novel (i.e., not previously noted) genera and families are indicated in bold in the table (Table 1). Visited plants in genera that have species that are native to the bee’s native regions of Mexico or Central America are identified with an asterisk. A comparison of these genera, with visited plants belonging to genera that lack native species in the bee’s native region, helped to address the question of whether this bee, like other introduced bees [39,40], prefers flowers from its native region.

## 3. Results and Discussion

### 3.1. General Findings

A total of 259 plant taxa (237 species and 22 horticultural forms), belonging to 156 genera and 56 families, included 263 total plant uses (Table 1). The majority were visited for floral rewards, totaling 248 plants belonging to 148 genera in 51 families. Visited for only the fragrance collection of volatile oils by males were 11 plants in 8 genera and 5 families. Of the 263 plant use observations, 196 were the author’s direct observations. Fifty-eight records were the author’s identifications of plants in photographs of *E. dilemma* visiting fifty-six flowers posted on iNaturalists and two on BugGuide (indicated as sources Table 1). Four observations were made by knowledgeable colleagues, two plant uses were recorded on museum specimen labels, and one was from the published literature (Table 1).

The 148 genera of plants in 51 families visited by *E. dilemma* for floral rewards in this study compares to 169 genera and 69 families visited by all ca. 200 Euglossine bees in Roubik and Hanson’s 2004 compilation of their own observations [6] and those of Zucchi et al. [26], Roubik [27], Kimsey [46], Dressler [4], Ackerman [28], Kress and Beach [29], and Ramirez et al. [7]. Despite the many plants recorded to be used by Euglossine bees in tropical America, very little documentation exists of what flowers are actually visited by most orchid bees where they normally live (David Roubik pers. com.) The large number of plants utilized by *E. dilemma* (259 taxa visited and 263 plant uses), a single bee in the relatively small geographical area of Florida, may have occurred for several reasons. First, these observations were made during an extended period of time, lasting almost 20 years. Second, Florida is quite rich horticulturally, with diverse plantings in residential areas, public gardens, and commercial nurseries, all of which were where the bee was observed. However, despite being widespread, the bee has a patchy occurrence. It is, for instance, absent in Fairchild Tropical Botanical Garden south of Miami, one of the world’s largest tropical gardens with an extensive assemblage of tropical plants. If and when *E. dilemma* spreads to that garden, it is likely to use more novel groups of uncommon tropical plants.

### 3.2. Plants Used and Novel Hosts

Of the 56 plant families, 20 were novel (Table 1 in bold) for the Euglossine bee tribe in all of the Neotropics, having taxa no previously recorded, even though no orchid bee is native to Florida. Floral resources were gathered from species in 16 of these families, including the Asparagaceae (genus *Lirope*), Asphodelaceae *(Dianella*), Berberidaceae (*Nandina*), Caryophyllaceae (*Dianthus*), Ebenaceae (*Disospryos*), Ericaceae (*Rhododendron*), Gelsemiaceae (*Gelsemium*), Iridaceae (*Iris*), Magnoliaceae (*Magnolia*), Nelumbonaceae (*Nelumbo*), Nyctaginaceae (*Bougainvillia*), Plumbaginaceae (*Pumbago*), Pontederiaceae (*Pontederia*), Scrophulariaceae (*Buddleja*), Surianaceae (*Suriana*), and Turneraceae (*Turnera*). Male bees gathered fragrances from the leaves of three other novel families: Vitaceae (*Parthenocissus*), Sapindaceae (*Litchi*), and Saururaceae (*Saururus*). The remaining novel family provided latex to females, Moraceae (*Ficus*).

The family with the most visited species was the Orchidaceae with 28 taxa (25 species and 3 horticultural forms) in 17 genera, 8 of which were novel. The other families with many visited species were Solanaceae with 27 taxa in 5 genera, including a single novel genus, Fabaceae, with 22 visited taxa in 13 genera, including 6 novel genera; Acanthaceae with 20 visited taxa in 12 genera, 11 of which were novel; Lamiaceae with 13 taxa in 4 genera, including 2 novel; Bignoniaceae with 13 taxa in 9 genera, including 2 novel; Asteraceae with 14 visited taxa in 11 genera, including 10 novel; Apocynaceae with 10 taxa in 9 genera, including 3 novel; Verbenaceae with 9 taxa in 4 genera, including 2 novel; Rubiaceae with 9 taxa in 8 genera, including 5 novel; Melastomataceae with 7 taxa in 6 genera, including 5 novel; and Passifloraceae with 5 taxa in 1 genus, which is not novel. The remaining 25 families have fewer than 5 visited taxa. Of the 156 genera of plants used by *E. dilemma*, about half, or 82, were novel, undocumented to have plants used by other orchid bees, and 76 of those included taxa visited for floral rewards.

The number of novel observations is noteworthy because it expands our understanding of the taxonomic groups that orchid bees can utilize. Flowers in 18/20 genera in the Acanthaceae were not previously recorded to be used by Euglossine bees. Most of these are ornamentals that are native to Asia, and they may be more common in Florida than in the horticultural floras of tropical America. The bee’s use of novel flowers of species in the Asteraceae is particularly interesting, because flowers in this family have been rarely reported to be used by orchid bees. In total, 9 of the 12 genera of these plants have not been previously reported to be used by *Euglossine* bees. Only *Cirsium* has been reported in Roubik and Hanson 2004. Also notable is that all 11 novel genera were identified from photographs posted on iNaturalist. I observed the bee visit the flowers of only two genera of Asteraceae. One, known as Mexican flame-vine, *Pseudogynoxys chenopoides* (Kunth) Crabrera (syn. *Senecio chenopodioides* Kunth), was almost always flowering in southern Florida, where *E. dilemma* was abundant, but I observed it to be visited only on one occasion. The other, *Cirsium horridulum* (Torr.&A.Gray) L.H.Baileya, is a frequently visited native plant in sunny terrestrial habitats in natural areas of the Everglades region. Photographs of the bee visiting *Bidens alba* DC., a native weed, were posted several times on iNaturalist (Table 1, iNaturalist 133788198, 133459837, and 68145475), but I never saw the bee use *B. alba* flowers, despite the plant being very abundant at many sites where the bee was abundant. Plants identified from iNaturalist posts constitute about 20% of the records but 37% of the novel genera of plants used by the bee, demonstrating the importance of this citizen science resource. *Euglossa dilemma*’s beautiful iridescent blue-green metallic coloration and its active flight behavior, such as hovering in front of flowers and human observers, make it quite apparent and attractive, which stimulates citizen scientists to document it. The bee’s high usage of novel plant hosts may be due to greater geographic coverage by these observers and the more temperate nature of the plants in central Florida. Both the native and non-native floras of Florida include a greater number of temperate plants, some of which Euglossines have not been previously exposed to in tropical America. It seems that the bee’s flower usage depends on what is available [47] and its ability to learn. For instance, if the nectar rich tubular flowers of *Hamelia patens* Standl., the most visited plant, are available, the bee may disregard the relatively small flowers of the Asteracaeae and many legumes. *Euglossa dilemma* used the red flowered ornamental herb *Pentas lanceolata* (Forssk) Deflers rarely, or almost never in some localities, but regularly in nearby localities.

### 3.3. Nativity of Plants Used

Regarding the question of whether Euglossine bees prefer plants from their native region, the majority of the species utilized by these bees in Florida have related species native to Mexico and central America. Of the 148 genera with species visited by the *E. dilemma* only for floral rewards, 113 (76%) have species that are native to the bee’s native region in tropical America.

### 3.4. Flower Usage- Nectar and Pollen Rewards

Flowers (247 taxa species and horticultural forms) were the most common plant part visited. Most (227) were visited to obtain nectar and or pollen. The flowers of 58 taxa, or about a fourth of the visited flowers, were visited exclusively by females to collect pollen because they lack nectar. Forty-two of these species have flowers with poricidal anthers and were buzz-pollinated when visited. These 42 species belong to 12 genera and 4 families and include 21 *Solanum* species (Solanaceae); 12 legumes in the subfamily Caesalpinioideae (*Cassia*, *Chamaecrista*, and *Senna*) (Fabaceae); 7 species in 6 genera in the Melastomataceae (*Centradenia*, *Dissotis*, *Heterocentron*, *Medinilla*, *Miconia*, and *Tetrazygia*); a single *Exacum* species in the Gentianacaeae, and *Dianella tasmanica* Hook.f, in the Asphodelaceae. The pollen-only flowers of the other 16 taxa lacking poricidal anthers include the male flowers of 6 *Begonia* taxa (Begoniaceae), 7 species in the Commelinaceae (*Commelina*, *Dichorisandra*, *Tradescantia*), and a male flower of the weedy dioecious cucurbit *Mormordica charantia*. In addition, pollen-only collection was also observed from the two erect exposed stamens of the half flowers of the so-called blue lips, *Sclerochiton harveyanus* Nees (Acanthaceae), and apparently from the brush flowers of guava *Psidium guajava* L. (Myrtacaeae). Pollen-only collections were frequently made from the red-orange flowers of the native/ornamental *Salvia coccinea* L., during which, the bees flew rapidly from flower to flower briefly collecting pollen from the tips of the exerted stamens. Nectar is also collected from these flowers but usually on separate foraging bouts.

Plant visitation primarily for floral nectar appears to have occurred in 120 taxa or about half of the flower species. In many cases, this was easy to see, because bees extend their very long tongues before flying into gullet flowers (those they completely enter) and before inserting them into the tubular flower to collect nectar. Without seeing whether or not the tongue is extended when a female bee enters a gullet flower, one cannot know what reward is being collected. Flowers in 43 plant taxa appeared to be visited for both nectar and pollen, which is about 20% percent of taxa that the bees visit. Male bees appeared to be more frequent visitors than females to the flowers of several *Stachytarpheta*: (*S. caatingensis* (Rich.) Vahl, S. *jamaicensis* (L.) Vahl, and *S. urticifolius* Sims; Verbenaceae).

### 3.5. Visits to Orchids and Floral Fragrances

Floral fragrances were collected by males from flowers of 14 species (6% of the taxa visited) belonging to 8 genera in 3 families. To collect the volatile oils that produce the fragrance, the bees land on the flower, locate the oils, secrete lipids from their mouth parts to dissolve the oils, sweep up the oils with brushes on their fore tarsi, and then hover to transfer the collected oils into their hind tibial storage containers. Orchids, mostly exposed as potted plants, constituted 12/14 of the plants visited for floral fragrance, and 8 were perfume orchids in the genera *Coryanthes, Gongora, Lycaste, Mormodes, and Stanhopea. Euglossa dilemma* was not observed to visit the two *Coryanthes* species, but were found dead within the bucket flowers of both species. One bee was male and the other a female.

Four species of orchids, which are not perfume orchids, were nevertheless visited by males to collect fragrance. *Cattleya quadricolor* Lindl. flowers were visited by many males that entered the flowers and collected fragrance from interior surfaces of the tubular labellum. The bees were too small to contact the column with the sexual parts so did not pollinate the flowers. Males also visited the flowers of *Myrmecophila tibicinis* (Bateman) Rolfe and collected fragrance from the external surfaces of the petals and sepals, but did not enter the flower so did not contact the column with the sexual parts. Similarly, males collected fragrances from exterior surfaces of *Aspasia epidendroides* Lindl., *Encyclia oncidioides* Schltr., and *Prosthechea radiata* (Lindl.) W.E.Higgins, but did not enter the flowers.

Of the orchids visited to collect volatile oils, nine were perfume orchids in the genera, *Aspasia, Coryanthes, Gongora, Lycaste, Mormodes, and Stanhopea*, all of which are pollinated by male orchid bees in tropical America [48]. *Euglossa dilemma* removed the pollinia from both *Lycaste* species and the single species of both *Gongora* and the *Mormodes*. If the pollinia of an orchid are removed by a visiting insect, it is considered to be a pollinator of the orchid [5,49]. This assertion seems valid because of the way that orchids manipulate their pollinators to place their pollinaria on specific sites on their bodies. The insects are then manipulated again to remove the pollinia when they visit another flower [4]. Ramirez et al.’s compilation of plants used by *E. viridissima* (prior to the separate *E. dilemma* from it) in tropical America [7] includes *Lycaste aromatica* Lindl. Although *E. dilemma* males collected fragrance from *Aspasia* and *Stanhopea* species, they did not remove pollinia, indicating that they may not be pollinators of these species. No *E. dilemma* visits were observed in the *Coryanthes* and the occurrence of the dead bees in the flowers will be discussed elsewhere. The common ornamental aroid *Spathiphyllum cannifolium* was visited by numerous males that collected fragrance oils during its few days of flowering. Male orchid bees are known to collect fragrance and pollinate *Spathiphyllum* species [17]. Male bees also collected fragrance the pineland heliotrope, *Euploca polyphylla* (Boraginacaeae), not previously known to be visited by male orchid bees for fragrance. No members of the Boraginaceae have been reported to have perfume flowers visited by Euglossine males, so further investigation is warranted.

The males’ fragrance-collecting behavior from the exterior surfaces of three orchids, *Aspasia epidendroides*, *Encyclia oncidioides* Schltr., and *Myrmecophila tibicinus* (Bateman) Rolfe, without entering the flowers, is not understood. Fragrance robbery of the flowers of *A. epidendriodes* by eglossine bees has been previously observed (James Ackerman, pers. com.).

Male orchid bees are known to collect fragrance from decaying plant material [13], and the flowers of *E. oncidioides* were old and dully colored so may have been infected with senescence fungi that attracted the bees. Similarly, where *E. dilemma* has recently colonized in the Dominican Republic [38], males have been observed collecting fragrance from orchid petals of *Tolumnia variegata* (Sw.) Braem (https://www.inaturalist.org/observations/6745735) (accessed on 10 February 2023), but apparently do not contact the column and thus do not effect pollination).

In total the flowers of 28 orchids, including 25 species and 3 horticultural forms, were visited by *E. dilemma*. None of these are native to Florida and most were exposed in potted plants in a garden setting where this bee was common. Nine are perfume orchids previously discussed; five species in the genera *Cattleya*, *Dendrobium* and *Sobralia* were visited to collect nectar. Another nine eleven species in *Arundina*, *Guarianthe*, *Phaius*, *Spathoglottis*, and *Vanilla* are deceptive orchids with flowers that resemble those that have nectar but have none. *Guarianthe skinneri* (Bateman)Dressler &W.E.Higgins, is a popular orchid grown outdoors in southern Florida, which is pollinated by *E. dilemma* [50]. Orchid flowers wilt a short time after they are pollinated, which makes *E. dilemma* pollination a potential threat to the commercial orchid industry of *Cattleya* and other orchids [51].

Flowers of 2/14 visited species for fragrance collection were not orchids. Male bees collected fragrance chemicals from the perfume flowers of the so-called peace lily, *Spathiphyllum canniaefolium* (Dryand.ex Sims) Schott (Araceae), a common ornamental. Male bees exhibited fragrance-collecting behavior when visiting the flowers of the native pineland heliotrope, *Euploca polyphylla* (Lehm.) J.I.M. Melo & Semir (Boraginaceae).

### 3.6. Floral Resin

Floral resin rewards were collected from the flowers of five species, including three *Clusia*, the native/ornamental tree *C. hilariana* Schltdl. and the ornamental shrubs *C. lanceolata* Cambess. and *C. orthoneura* Standl. (Clusiaceae). The other two were *Dalechampia* vines (Euphorbiaceae). *Dalechampia aristolochifolia* Kunth is an ornamental with a large pink bract subtending small flowers, and *D. scandens* L. is a weed with a large greenish bract subtending and partially enclosing small flowers.

The resin reward flowers of the three *Clusia* and two *Dalechampia* species are in groups that have evolved Euglossine bee pollination [21], by female bees that collect the malleable resins to fashion their brood cells and build their entire nests in some cases [52]. These floral resins are also collected by meliponine bees (David Roubik, pers. com). *Dalechampia scandens* L. is a recently naturalized vine with stinging hairs and an inflorescence of small apetalous flowers with a resin gland, subtended by a large green leaf-like bract [53]*. Euglossa dilemma* is a pollinator of this plant in its native Mexico and the only known pollinator in Florida [54]. *Clusia hilariana* is a native Neotropical tree composed of only female plants with apomictic flowers in Florida. *Clusia lanceolata* is an ornamental shrub native to Brazil. *Clusia* flowers secrete resin in the centers of the flowers in a circle surrounding the sexual parts. When potted *C. lanceolata* plants were added to the garden, where *E. dilemma* was abundant, female bees visited the flowers and began collecting resin within an hour of their placement. The plant is uncommon in Florida and the bees were unlikely to have had previously encountered it in the area of study.

### 3.7. Non-Floral Resources

Eleven plants in eight genera and eight families have plant parts from which male bees collected fragrance. Ten of these were leaves and five were frequently visited often by multiple bees: four basils (3 *Ocimum* species and one variety) (Lamiaceae) and allspice (*Pimenta dioica* (L.) Merr. (Myrtaceae). Before and during the collection from these strongly attractive plants, the bees chewed the central veins and leaf margins, apparently to better access the desired chemicals. The leaves of the five other plants were rarely observed to be visited by the bees. Two of these plants, the native emergent aquatic wetland herb *Saururus cernuus* L. (Saururaceae), and the invasive tree *Melaleuca quinquenervia* (Cav.) S.T. Blake (Myrtaceae), have aromatic leaves. The other three, *Parthenocissus quinquefolia* (L.) Planch. (Vitaceae), a native vine, the invasive shrub *Solanum diphyllum* Kunth (Solanaceae), and the fruit tree *Litchi chinensis* Sonn. (Sapindaceae), have leaves that are not noticeably aromatic and may have been infected by microorganisms apparently creating aromatic chemicals attractive to the bees. The remaining nonfloral plant source was the native/ornamental tree gumbo limbo (*Bursera simaruba* Sarg. Burseraceae). Multiple bees exhibited fragrance-collecting behavior on a section of the tree trunk below the holes of a buprestid beetle borer where sawdust-like borings of the beetles had accumulated. The dried flowers of a goldenrod, *Solidago* sp. (Asteraceae) were visited by males to collect fragrance chemicals.

Female bees collected resin from a broken twig of the tree *Clusia hilariana*, which they use to construct brood cells and to seal their nests. A female bee was also identified collecting latex from the stem of a *Ficus* sapling (Moraceae), also probably to use in her nest.

Four basils (three *Ocimum* species and one variety) and allspice (*Pimenta dioica*) were especially attractive to the male bees and all showed chewing damage to the leaf margins and to the central veins of allspice leaves. Basil leaves could be torn to attract the male bees, but their attractiveness diminished a few hours after leaves were torn. The leaves of the other four plants rarely were observed attracting bees for fragrance. *Melaleuca quinquenervia* (Cav.) S.T.Blake and *Saururus cernuus* L. have distinctively scented leaves, so the bees were probably seeking chemicals made by the plants. The other two plants, *Solanum diphyllum* L. and *Litchi chinensis* Sonn., which have unscented leaves, may have been infected by microorganisms, causing them to produce chemicals that attracted bees. The males collecting from the trunk of a gumbo limbo, *Bursera simaruba*, that had been bored by buprestid beetles, were probably obtaining aromatic wound compounds from resin. Male orchid bees have been reported to collect resin from *Bursera* species [21]. The dried flowers of a *Solidago* (Asteraceae) visited by males to collect fragrance compounds which may been senescence chemicals rather than floral rewards. There are four iNaturalist photographs of *E. dilemma* males collecting from such *Solidago* flowers on different dates in two years and locations (41). As an interesting aside, male *E. dilemma* have been observed to collect the herbicide triclopyr 2-butoxyethyl ester [24], which was sprayed on the invasive weed, Brazilian pepper (*Schinus terebenthifolia* Raddi).

### 3.8. Flower Form and Rewards

Gullet flowers, such as those of *Jacaranda* (Bignoniaceae), are those that bees crawl inside to. These were the most common flower form used by this bee, occurring in 70 taxa or 29% of the flowers. Gullet flowers included all 13 species in the Bignoniaceae, many orchids, and in 12 Acanthaceae, 8 Apocynaceae, and all 5 Costaceae.

Tubular flowers, such as those of *Hamelia* species (Rubiacacee), were found in 55 taxa or 22% of the flowers and were the second most used flower form. Nectar rewards in tubular flowers and were accessible to *E. dilemma* because of its long tongue. Tubular flowers occurred in all 10 taxa of Lamiaceae and 9 in the Verbenaceae, and in 8 of 9 Rubiaceae, 7 Acanthaceae, all 4 Lythraceae, and all 4 Bromeliaceae.

There were 42 taxa with flowers having poricidal anthers that were buzz pollinated, and they accounted for 16.6% of the taxa utilized by the bee. In buzz pollination, the bees typically grasp the conical stamen with their claws or mandibles and then vibrate their flight muscles at a frequency that causes the small light pollen grains to vibrate out from pores at the distal tips of the anthers. This usually causes a high-pitched audible sound to the human ear. Pollen from buzz pollination is typically from pollen-only flowers and is delivered to the stigma during the buzzing. Sometimes, a bee pollinates the flowers with both self-pollen and conspecific pollen. These included all 21 *Solanum* species (Solanaceae), 12 of 22 Fabaceae, and all 7 Melastomataceae. The remaining 75 floral taxa were both phylogenetically and morphologically diverse. I classified flowers of 60 of these taxa as “open” if their rewards were exposed and easily accessed. Among these were the 7 Commelinaceae and 6 *Begonia* visited for pollen, and the 14 Asteraceae with small florets. Twelve flowers were categorized as “closed” because their resources were not readily available to most insect visitors. These flowers required manipulation as seen in the six papilionaeous legumes (known bee flowers); in accessing the nectar beneath the floral coronas in the five *Passiflora* spp.; and in finding the hidden nectar of the single *Iris*.

Regarding the form of the flowers and reward accessibility, 180/247 (72.4%) of the flowers visited by *E. dilemma* have their rewards protected in gullet flowers, tubular flowers, flowers with poricidal anthers, in papilionaceous flowers, and beneath coronas in the flowers of *Passiflora*. If the 25 species of orchid visited for their actual or apparent rewards, this figure increases to 81.2% (202/247) of the flowers. The bee’s use of tubular flowers is significant, because many of these flowers are more commonly visited by butterflies and moths in Florida.

The exceptionally long tongue of *E. dilemma* allows it to access the rewards in tubular and gullet flowers. At rest the tongue is held beneath the 11–12 mm long body of the bee and it extends to almost the end of the body. The bee appears to have the longest tongue of any bee in Florida. It can access quite restricted floral resources, is a part of diverse food webs, and potentially pollinates more flowers. The number of nectar hosts visited by a specific species of orchid bees increased significantly with tongue length [47]. In a study on the pollination of the tubular flowers of the native tropical shrub *Guettarda scabra* (L.) Vent. (Rubiaceae) in southern Florida, *E. dilemma* was the only bee visiting the flowers [55]. Neotropical flowers, including the three *Costus* species used by *E. dilemma* in Florida, have evolved gullet flowers for orchid bee pollination [47,56].

### 3.9. Flower Color

Of 201 plants with uniformly colored flowers, the most common colors were yellow (43 taxa) and white with (44 taxa), followed by pink with 36 and blue with 27. The next most common colors of visited flowers were rose with 18 and purple with 16. There were only 7 orange and 6 red flowers. There were 30 multicolored flowers.

*Euglossa dilemma* visited flowers of many different colors, but the colors were not scientifically defined so this information cannot be meaningfully compared to that of other studies. Color vision of *E. dilemma* males has peaks in the ultraviolet, blue and green regions, indicating that their vision is similar to other corbiculate (pollen-basket-, i.e., stingless bees, honeybees, and bumblebees) [57]. Nevertheless, only seven of the flowers that the bee visited (Table 1) were red to the human eye, including *Pentas lanceolata* (Rubiaceae), one of the most visited flowers by *E. dilemma* in Florida, due in part to its ubiquitous presence as an ornamental.

### 3.10. Life Forms and Plant Types

Herbaceous plants were the most visited life form with 133 of/259 taxa (52%), followed by shrubs 65/259 (24%), vines 36/259 (14.4%), and trees 27/259 (10.4%). Of the 259 taxa, ornamental plants were the most common type of plants used with 196 taxa (76%). Of the ornamentals 180 taxa (92.6%) were non-native, 22 had naturalized in the region, and 7 were invasive. *Euglossa dilemma* utilized 55 species of plants native to Florida, which represents 20.4% of the total. A total of 16 human food or spice plants were visited, 12 of which were introduced and 4 were native. Plants used for food included eggplant and tomato (*Solanum*), orange (*Citrus x sinensis* (L.) Osbeck), red pepper (*Capsicum annuum* L.), the vanilla orchid (*Vanilla planifolia* Andrews), and black sapote (*Disospryos digyna* Jacq). Seventeen visited plants (6.8%) were weeds, all introduced, and ten (4%) were invasive, including seven escaped ornamentals.

A large number and wide range of plant types are visited by *E. dilemma*, but because visitation frequencies were not recorded and pollination research was not part of this study, the importance of the bee to the visited plants, and vice versa, is unknown. Exceptions regarding pollination are the *Solanum* and *Senna* species, which are often pollinated when they are buzzed [22]. Plants in both genera were intensively used by the bee. When potted *Solanum* of less common species were presented in the garden setting, they were quickly and intensively used. These *Solanum* species illustrate the diverse plant types used by *E. dilemma*, which included important food plants (tomato and eggplant), ornamentals (*S. wrightii* Benth.), invasive weeds (*S. viarum* Dunal and *S. torvum* Sw.), and natives (*S. bahamense* Mill. and *S. donianum* Walp.). One study found *E. dilemma* is the most important pollinator of the invasive weed *S. torvum* in southern Florida [58].

An analysis of nest provisions of *E. dilemma* and E. *viridissima* in Mexico found that 88% of the pollen volume in the nests was comprised of *Solanum*, *Senna*, and *Ocimum* species [31,33]. Arriaga and Hernández [30] found that the most important pollen in the provisions of *Euglossa atroveneta* Dressler in Mexico included species of *Cassia*, *Commelina*, *Lycianthes*, *Solanum*, and *Tibouchina*. Species in all these genera were utilized by *E. dilemma* in Florida. The value of buzz pollination to tomatoes can be significant. A meta-analysis of the buzz pollination of tomatoes found a significantly increased fruit weight compared to the controls [59]. The observations of the bee’s visits to *Citrus* sp. and eggplant (*Solanum melongena* L.) were too few to know its potential pollination significance. *Thalia geniculata* L. (Marantaceae) is an emergent native wetland herb visited by *E. dilemma*. It has flowers have mobile anthers that display explosive pollen dispersal when touched by visiting euglossine bees in Costa Rica [60,61].

### 3.11. Most Common Bee Resources

Although data were not collected on the frequency of visitation to particular plants, the bee’s regular visits to some flowers was apparent. The widely occurring and long-flowering, native/ornamental rubiaceous shrub, *Hamelia patens*, is probably the most visited plant in Florida for nectar and apparently also for pollen almost all year. Of 498 iNaturalist photos of *E. dilemma* visiting flowers [41], 101 or ca. 20% are of *H. patens. Stachytarpheta* species, particularly the native *S. jamaicensis* (L.) Vahl and the naturalized *S. urticifolius* Sims, also are flowers that are very often used for both nectar and pollen. Many of the common buzz-pollinated native and ornamental *Senna* and *Solanum* species are visited daily their pollen (no nectar is produced), particularly in early morning. When *Tabebuia* and *Jacaranda* trees and *Tecoma stans* (L.) Juss. ex Kunth shrubs are flowering, *E. dilemma* visits these gullet flowers to collect nectar daily during most, if not all of their flowering periods. The ornamental herb *Ruellia simplex* C. Wright (Acanthaceae), called the Mexican petunia, is also heavily used, mostly for nectar. The resin reward flowers of the ornamental *Clusia lanceolata* and the newly naturalized weed *Dalechampia scandens* are frequently visited for their high-value resin, although both species have, at present, limited distributions in Florida. Since *E. dilemma* has spread into Everglades National Park, it can be readily seen on the flowers of the native thistle *Cirsium horridulum* during the winter and spring. The pollen-only flowers of *Commelina erecta* L., a native herb common on coastal strand, are frequently visited during the morning hours during much of the year. Another frequently visited beach plant is the pea *Canavalia rosea* (Sw.) DC, used for nectar and pollen. Most of these plants and species belonging to all 14 of these genera occur in Mexico and Central America, sympatrically with *E. dilemma*.

### 3.12. Comparisons of Resources Used Elsewhere

The flowers visited by *E. dilemma* in Florida are similar to those found in other recent research. Ferreia-Caliman et al. [34] studied the food resources of *Euglossa cordata* L. in an urban area in southeastern Brazil. They found *Dalechampia stipulacea* Mull.Agr. (Euphorbiaceae) acted as a floral resin source. Pollen sources included species from seven families; five of them were plants with poricidal anthers, including those in the Commelinaceae, Fabaceae, Melastomataceae, and Solanaceae, which are used by *E. dilemma* in Florida. Nectar was collected primarily from plants with long, tubular corollas such as Acanthaceae, Apocynaceae, Bignoniaceae, and Convolvulaceae, all frequently used by *E. dilemma* in Florida.

Rocha-Filho et al. [31] studied the plants used by 14 species of Euglossine bees in three genera (*Eufriesea*, *Euglossa*, and *Eulaema*) in the coastal Atlantic Forest of Brazil and recorded 105 species of plants. There is broad similarity in the plants used by these bees and those used by *E. dilemma* in Florida. The main difference is that the Brazilian bees collected pollen from nine species in the Myrtacaeae, while in Florida only guava was used in that family. The native Myrtaceae exists primarily in remnants of tropical hardwood forests in subtropical southern Florida, thus may have been overlooked. The Myrtaceae in Florida include *Melaleuca quinquenervia*, a highly invasive tree in southern Florida, which has been subject to eradication efforts but is still present in natural areas. The degree of common usage between the 14 Brazilian bees and *E. dilemma* is striking, especially since more bee species were involved in the Brazilian study, including larger bodied *Eufriesea* and *Eulaema* species.

### 3.13. Invasive Bee Resource Choices

Introduced or invasive bees may pollinate floral resources that have also been introduced [40]. For instance, in Australia and New Zealand, introduced honeybees and bumblebees are the primary pollinators of introduced weeds [40]. In North America, the naturalized giant resin bee *Megachile sculpturalis* Smith, may preferentially visit the flowers of plants that have been introduced from its native range in Asia [62]. Most flowers that it uses are introduced plants from Asia [63]. *Euglossa dilemma* seems to follow this pattern. Among Floridian flowering plant genera visited by the bee for floral rewards, 113/148 (76%) are native to tropical America, suggesting a preference for the plants of this region. Moreover, the 14 most visited genera have species that are native where the bee occurs naturally in Mexico and Central America. A significant exception is *E. dilemma*’s frequent use of pollen from non-native plants: seven of 12 plant genera buzz collected by the bee are non-native. This contrasts markedly with *Euglossa cordata* L. in Brazil, which uses only native plants for pollen [34].

The foraging preferences or choices of any bee are influenced by the relative abundance and accessibility of local plants or flowers. The apparent preference of *E. dilemma* for plants from its native region might occur if Neotropical ornamental plants exist in greater abundance than ornamentals from other regions of the world, or even, in some cases, being more abundant than plants native to Florida. In addition, the native floras of Florida, particularly southern Florida, and the Neotropics are biogeographically and botanically related. It is rather striking that the majority of the most visited plant species in Florida are themselves native to the Neotropics.

### 3.14. Predictions of Plant Usage

Using these extensive resource use data, the following predictions were made. (1) The bee’s presence in Florida, distant from its native region of Mexico and Central America and the geographical ranges of other orchid bees, results in the usage of many new taxa of plants. As observed, half, 82/148 (76%), of the genera and one third, 16/51(31%), of the plant families with floral resources used by the bee, were not previously recorded as utilized by Euglossine bees. (2) Like other naturalized bees, it used relatively more plants from its native range. As recorded, 113/148 (76%) of genera with species bearing collected floral rewards are native to or congeneric with plants native to the bee’s native region. Many of the novel plants or their congeners utilized by *E. dilemma* are native to the bee’s native range, but not previously documented to be used there. (3) Given the bee’s long tongue, ability to buzz pollen from poricidal anthers, and ability to collect and use specialized rewards, it would disproportionately use taxa with protected or highly specialized floral rewards. As demonstrated, 180/247 (72%) utilized plants had flowers that bear rewards which are protected and unavailable to, or of no interest to, most other flower visitors.

**Table 1 insects-14-00909-t001:** **Resources collected by *Euglossa dilemma* in Florida.** Bolding of names = novel genus and family for Euglossine bees. Aterisk * indicates the genus is sympatric with the bee in Mexico and Central America. Flower form and reward access: ^1^ gullet-large mouthed flower the bee can enter, ^2^ tubular-long-tongued bees can access rewards, ^3^ open-rewards easily accessed, ^4^ buzz-pollen is vibrated out, ^5^ closed-flowers require manipulation to access rewards.

Plant Family Plant Species		Plant Habit	Plant Type	Collectable Product (Sex)	Flower Color	Flower Form ^1, 2, 3, 4, 5^		Source & Notes
Acanthaceae								
	*Asystasia gangetica*	Herb	Naturalized ornamental	Nectar	Pink	Gullet ^1^		Author
	*Asystasia travancorica*	Herb	Ornamental	Nectar	Rose	Gullet ^1^		Author
	***Barleria*** *cristata*	Herb	Ornamental	Nectar	White and blue	Gullet ^1^		Author
	***Barleria*** *oenotheroides*	Shrub	Ornamental	Nectar	Yellow	Gullet ^1^		Author
	***Eranthemum*** *pulchellum*	Shrub	Ornamental	Nectar	Blue	Tubular ^2^		Author
	* ***Justicia*** *brandegeeana* X	Shrub	Ornamental	Nectar	White and rose	Tubular ^2^		Author
	* ***Justicia*** sp.	Shrub	Ornamental	Nectar	Yellow and orange	Tubular ^2^		Author
	* ***Odontonema*** *callistachyum*	Shrub	Ornamental	Nectar	Lavender	Tubular ^2^		Author
	* ***Odontonema*** *cuspidatum*	Shrub	Ornamental	Nectar	Red	Tubular ^2^		Author
	* ***Pachystachys*** *lutea*	Shrub	Ornamental	Nectar	White	Tubular ^2^		Author
	* ***Pseuderanthemum***	Shrub	Ornamental	Nectar	Pink and	Tubular ^2^		Author
	*laxiflorum*				lavender			
	* *Ruellia brittoniana*	Herb	Invasive/	Nectar	Purple	Gullet ^1^		Author
			ornamental					
	* *Ruellia caroliniensis*	Herb	Native	Nectar	Purple	Gullet ^1^		Author
	* *Ruellia elegans*	Herb	Ornamental	Nectar	Red	Gullet ^1^		Author
	* ***Sclerochiton*** *harveyanus*	Herb	Ornamental	Pollen	Blue	Open ^3^		Author
	***Strobilanthus*** *flaccidifolius*	Herb	Ornamental	Nectar	Maroon	Gullet ^1^		Author
	***Strobilanthus*** sp. 1	Herb	Ornamental	Nectar	Rose	Gullet ^1^		Author
	***Strobilanthus*** poss. sp. 2	Herb	Ornamental	Nectar	Blue	Gullet ^1^		Author
	***Suessenguthia** multisetosa*	Vine	Ornamental	Nectar	Pink	Gullet ^1^		Author
	*Thunbergia erecta*	Shrub	Ornamental	Nectar	Purple	Gullet ^1^		Author
Apocynaceae								Author
	* *Allamanda schottii*	Shrub	Ornamental	Nectar/pollen	Yellow	Gullet ^1^		
	* ***Asclepias*** *curassavica*	Herb	Naturalized ornamental	Nectar	Red and orange	Open ^3^		Author
	* ***Asclepias*** *tuberosa*	Herb	Native	Nectar	Orange	Open ^3^		BugGuide NicoleTharp
	* *Cascabela thevetia*	Shrub	Ornamental	Nectar/pollen	Yellow	Gullet ^1^		iNaturalist z7nikon
	* *Cascabela* sp.	Tree	Ornamental	Nectar/pollen	Yellow	Gullet ^1^		Author/*Thevetia*
								*peruviana*
	*Catharanthus roseus*	Herb	Naturalized/ ornamental	Nectar/pollen	Pink	Tubular ^2^		Author/*Thevetia* sp.
	* *Mandevilla* sp.	Herb	Ornamental	Nectar	Rose	Gullet ^1^		iNat Logan Crees
		Shrub	Ornamental	Nectar/pollen	Rose	Gullet ^1^		Author
	* ***Pentalion*** *luteum*	Vine	Native/	Nectar	Yellow	Gullet ^1^		Author
			ornamental					
	* *Tabernaemontana divaricate*	Tree	Ornamental	Nectar/pollen	White	Gullet ^1^		Author
Araceae								
	*Spathiphyllum canniaefolium*	Herb	Ornamental	Fragrance M	NA	Open ^3^		Author/manybees
**Asparagaceae**								
	***Liriope*** *muscari*	Herb	Ornamental	Nectar?	Lavender	Open ^3^		iNat AidanMarshal
**Asphodelaceae**								
	***Dianella*** *tasmanica*	Herb	Ornamental	Pollen	White and blue	Buzz ^4^		iNat 118044092
Asteraceae	* ***Bidens*** *alba*	Herb	Native/weed	Nectar	White	Open ^3^		iNat ArthurWindsor
	* *Cirsium horridulum*	Herb	Native	Nectar/pollen	Pink	Open ^3^		Author
	* *Cirsium nuttallii*	Herb	Native	Nectar/pollen	Pink	Open ^3^		Roger Hammer
	* ***Conoclinum*** *coelestinnum*	Herb	Native/	Nectar	Blue	Open ^3^		Author
			ornamental					
	* ***Cosmos*** *bipinnatus*	Herb	Ornamental	Nectar	Pink	Open ^3^		iNat bichir
	* ***Helianthus*** *annuus*	Herb	Ornamental	Nectar	Yellow and	Open ^3^		iNat hjitrapp
					brown			
	* ***Helianthus*** *debilis*	Herb	Native/	Nectar	Yellow and	Open ^3^		iNat quester
			ornamental		black			
	* ***Liatris*** *prob. garberi*	Herb	Native	Nectar	Pink	Open		iNat Betina
								Colley
	* ***Pityopsis*** *graminifolia*	Herb	Native	Nectar	Yellow	Open ^3^		iNat Joe MDO
	* ***Pseudogynoxys***	Vine	Ornamental	Nectar	Orange and	Open ^3^		iNat michael-
	*Chenopoides*				yellow			Ofthepines
	* ***Solidago*** sp.	Herb	Native	Fragrance M	NA	Open ^3^		Author/old flowers
	* ***Tithonia*** *rotundifolia*	Herb	Ornamental	Nectar	Orange and	Open ^3^		iNat aliandbrice
					yellow			
	* *Veronia* sp.	Herb	Native/	Nectar/pollen	Fushia	Open ^3^		iNat Yolanda
			ornamental					Svatik
	* ***Zinnia*** *elegans*	Herb	Ornamental	Nectar	Orange	Open ^3^		iNat Sdickman
Balsaminacae								
	* *Impatiens balsamina*	Herb	Ornamental	Nectar	Rose	Gullet ^1^		iNat Yolanda
								Svatik
	* *Impatiens walleriana*	Herb	Ornamental	Nectar	Rose	Tubular ^2^		iNat abaum
Begoniaceae								
	* *Begonia coccinea*	Herb	Ornamental	Pollen	Pink	Open ^3^		Author
	* *Begonia obliqua*	Herb	Ornamental	Pollen	White	Open ^3^		Author
	* *Begonia* Pig Skin	Herb	Ornamental	Pollen	White	Open ^3^		Author/*B. odorata*
	* *Begonia* Top Hat	Herb	Ornamental	Pollen	White	Open ^3^		Author
	* *Begonia* cultivar 1	Herb	Ornamental	Pollen	White	Open ^3^		Author
	* *Begonia* cultivar 2	Herb	Onamental	Pollen	White	Open ^3^		Author
**Berberidaceae**								
	***Nandina*** *domestica*	shrub	Invasive/	Pollen	White	Open ^3^		iNat craigdux
			ornamental					
Bignoniacaeae								
	* *Bignonia magnifica*	shrub	Ornamental	Nectar	Rose	Gullet ^1^		Author
	* *Dolichandra unguis-cati*	Vine	Naturalized/ ornamental	Nectar	Yellow	Gullet ^1^		Chris Howell/
								*Macfadyena unguis-cati*
	* *Jacaranda caerulea*	Tree	Ornamental	Nectar	Blue	Gullet ^1^		Author
	* *Jacaranda jasminoides*	Tree	Ornamental	Nectar	Purple	Gullet ^1^		Author
	* *Handranthus impetigenosa*	Tree	Ornamental	Nectar	Rose	Gullet ^1^		Author
	* *Mansoa alliacea*	Vine	Ornamental	Nectar	Violet	Gullet ^1^		Author/*Tabebuia*
								*impetigenosa*
	***Pandorea*** *jasminoides*	Vine	Ornamental	Nectar	Pink	Gullet ^1^		Author
	***Radermachera*** *gigantea*	Tree	Ornamental	Nectar	Pink and	Gullet ^1^		Author
					orange			
	***Radermachera*** *yunnanensis*	Shrub	Ornamental	Nectar	White and	Gullet ^1^		Author
					yellow			
	* *Tabebuia aurea*	Tree	Ornamental	Nectar	Yellow	Gullet ^1^		Author
	* *Tabebuia chrysotricha*	Tree	Ornamental	Nectar	Yellow	Gullet ^1^		Author
	* *Tabebuia heterophylla*	Tree	Naturalized/	Nectar	Pink	Gullet ^1^		Author
			ornamental					
	* *Tecoma stans*	Shrub	Naturalized/	Nectar/pollen	Yellow	Gullet		Author
			ornamental					
Boraginaceae								
	* ***Euploca*** *polyphylla*	Herb	Native	Fragrance M	Yellow	Tubular ^2^		Author/*Heliotropium*
								*leavenworthii*
Bromeliaceae								
	* *Aechmea gamosepala*	Herb	Ornamental	Nectar	Purple and	Tubular ^2^		Author
					pink			
	* *Aechmea* Blue Tango	Herb	Ornamental	Nectar	Blue and pink	Tubular ^2^		iNat sandrae-
								34242
	* *Aechmea* cultivar	Herb	Ornamental	Nectar	Yellow	Tubular ^2^		iNat jessa-Second
	* ***Wallisia*** *cyanea*	Herb	Ornamental	Nectar	Purple	Tubular ^2^		iNat allimeus-421/*T. cynea*
Burseraceae								
	* *Bursera simaruba*	Tree	Native/	Fragrance M	NA	NA		Author/ex tree bark
			ornamental					
Cannaceae								
	* *Canna indica*	Herb	Ornamental	Nectar	Yellow	Tubular ^2^		Author
**Caryophilaceae**								
	***Dianthus*** *gratianopolitanus*	Herb	Ornamental	Nectar	Pink	Tubular ^2^		iNat Laura Helm
Clusiaceae								
	* *Clusia lanceolata*	Shrub	Ornamental	Floral resin F	White and	Open ^3^		Author
					maroon			
	* *Clusia orthoneura*	Shrub	Ornamental	Floral resin F	Rose	Open ^3^		Author
	* *Clusia hilariana*	Tree	Native/	Floral resin F	White and	Open ^3^		iNat pufferfish
			ornamental		maroon			
	* *Clusia hilariana*	Tree	Native/	Wound resin F	NA	NA		Author/*C. rosea* ex twig
			ornamental					
Combretaceae								
	* *Combretum latifolium*	Vine	Ornamental	nectar/pollen	Orange	Brush		Author
Commelinaceae								
	* *Commelina benghalensis*	Herb	Weed	Pollen	Blue	Open ^3^		Author/*C. aubletii*
	* *Commelina errecta*	Herb	Native	Pollen	Blue	Open ^3^		Author
	* *Dichorisandra penduliflora*	Herb	Ornamental	Pollen	Blue	Open ^3^		Author
	* *Dichorisandra thyrsiflora*	Shrub	Ornamental	Pollen	Blue	Open ^3^		Author
	* ***Tradescantia*** *pallida*	Herb	Naturalized/	Pollen	Blue	Open ^3^		Author
			ornamental					
	* ***Tradescantia*** *roseolens*	Herb	Native	Pollen	Blue	Open ^3^		iNat Valerie Anderson
	* ***Tradescantia*** *spathacea*	Herb	Ornamental	Pollen	Blue	Open ^3^		Author
Convolvulaceae								
	* *Ipomoea carnea*	Vine	Ornamental	Nectar/pollen	White	Gullet ^1^		Author
	* *Ipomoea indica*	Vine	Native	Nectar/pollen	Purple	Gullet ^1^		Author
	* *Ipomoea pes-caprae*	Vine	Native	Nectar/pollen	Pink	Gullet		Author
	* *Ipomoea triloba*	Vine	Weed	Nectar/pollen	Pink	Gullet ^1^		Author
	* ***Merremia*** *tuberosa*	Vine	Naturalized/	Nectar/pollen	Yellow	Gullet ^1^		iNat lisnel
			ornamental					
Costaceae								
	* *Costus barbatus*	Herb	Ornamental	Nectar	Yellow	Gullet ^1^		Author
	* *Costus dubius*	Herb	Ornamental	Nectar	Yellow	Gullet ^1^		Author
	* *Costus malortieanus*	Herb	Ornamental	Nectar	Yellow and	Gullet ^1^		Author
					maroon			
	* *Hellenia speciosus*	Herb	Ornamental	Nectar	White	Gullet ^1^		AMNH _BEE252041/
								*Costus speciosa*
	* ***Monocostus*** *uniflorus*	Herb	Ornamental	Nectar	Yellow	Gullet ^1^		Author
Cucubitaceae								
	* *Cucurbita pepo* female	Vine	Food—vegetable	Nectar	Yellow	Gullet ^1^		Author
	* *Cucurbita pepo*	Vine	Food—vegetable	Fragrance? M	Yellow	Gullet ^1^		Author
	*Momordica charantia* male	Vine	Weed	Pollen	Yellow	Open ^3^		Bugguide Brice
**Ebenaceae**								
	* ***Disospryos*** *digyna*	Tree	Food—fruit	Nectar	Green	Open ^3^		Author
**Ericaceae**								
	* ***Rhododendron*** sp.	Shrub	Ornamental	Nectar	Rose	Gullet ^1^		iNat obrock
Euphorbiaceae								
	* *Dalechampia aristolochiifolia*	Vine	Ornamental	Resin F	Pink	Open ^3^		Author
	* *Dalechampia scandens*	Vine	Weed	Resin F	Green	Open ^3^		Author
Fabaceae								
	* ***Chapmannia*** *floridana*	Herb	Native	Nectar	Yellow	Open ^3^		iNat Jeff Weber
	* ***Bauhinia*** blakeana X	Tree	Ornamental	Nectar	Rose	Open ^3^		Author
	* ***Calliandra***	Shrub	Ornamental	Nectar	Rose	Open ^3^		Author
	*haematocephala*							
	* *Canavalia rosea*	Vine	Native	Nectar/pollen	Pink	Open ^3^		H. Flores
	* *Cassia fistula*	Tree	Ornamental	Pollen	Yellow	Buzz ^4^		Author
	* *Centrosema virginianum*	Vine	Native	Pollen	Pink	Closed ^5^		Author
	* *Chamaecrista fasciculata*	Herb	Native	Pollen	Yellow	Buzz ^4^		iNat gloria-
								markiewicz
	* *Clitoria ternatea*	Vine	Ornamental	Nectar	Blue	Closed ^5^		Author
	* *Crotalaria* poss.	Herb	Ornamental	Nectar/pollen	Yellow	Closed ^5^		Author
	*laburnifolia*							
	* ***Macroptilium*** *lathyroides*	Herb	Weed	Nectar	Maroon	Closed ^5^		Author
	* *Mimosa strigillosa*	Herb	Native/	Pollen	Rose	Closed ^5^		iNat dennis-
			ornamental					vonlinden
	* *Senna alata*	Shrub	Naturalized/	Pollen	Yellow	Buzz ^4^		iNat Laura
			ornamental					Zurro
	* *Senna didymobotrya*	Shrub	Ornamental	Pollen	Yellow	Buzz ^4^		Author
	* *Senna ligustrina*	Shrub	Native	Pollen	Yellow	Buzz ^4^		Author
	* *Senna mexicana*	Shrub	Native	Pollen	Yellow	Buzz ^4^		Author
	* *Senna nitida*	Tree	Ornamental	Pollen	Yellow	Buzz ^4^		Author
	* *Senna occidentalis*	Herb	Weed	Pollen	Yellow	Buzz ^4^		Author
	* *Senna pendula*	Shrub	Naturalized/	Pollen	Yellow	Buzz ^4^		Author
			ornamental					
	* *Senna polyphylla*	Tree	Ornamental	Pollen	Yellow	Buzz ^4^		Author
	* *Senna surattensis*	Tree	Naturalized/	Pollen	Yellow	Buzz ^4^		Author
			ornamental					
	* *Senna* sp.	Shrub	Ornamental	Pollen	Yellow	Buzz ^4^		Author
	* ***Sophora*** *tomentosa*	Shrub	Native/	Nectar	Yellow	Closed ^5^		Author
			ornamental					
**Gelsemiaceae**								
	* ***Gelsemium*** *sempervirens*	Vine	Native/	Nectar?	Yellow	Gullet ^1^		iNat dekaff
			ornamental					
Gentianaceae								
	***Exacum*** *affine*	Herb	Ornamental	Pollen	Blue	Buzz ^4^		Author
**Iridaceae**								
	* ***Iris*** *savannarum*	Herb	Native	Nectar	Blue	Closed ^5^		iNat Danile Onea
Lamiaceae								
	***Plectranthus*** *amboinicus*	Herb	Food—spice	Nectar/pollen	Blue	Tubular ^2^		iNat sunraythewingman
								*Coleus amboinicus*
	***Lavandula*** *angustifolia*	Herb	Ornamental	Nectar/pollen	Pink	Tubular ^2^		Author
	***Lavandula*** *multifida*	Herb	Ornamental	Nectar	Pink	Tubular ^2^		Author
	* *Ocimum basilicum*	Herb	Food—herb	Nectar	White	Tubular ^2^		iNat reesedel-
								rossobiodiversity
	* *Ocimum basilicum*	Herb	Food—herb	Leaf chem M	N/A	N/A		Author
	* *Ocimum basilicum*	Herb	Food—herb	Leaf chem M	N/A	N/A		iNat kat_jones
	*Minimum*							
	* *Ocimum gratissimum*	Herb	Ornamental	Leaf chem M	N/A	N/A		Author
	* *Ocimum sanctum*	Herb	Food—herb	Leaf chem M	N/A	N/A		Author
	* *Salvia coccinea*	Herb	Native/	Nectar/pollen	White, pink,	Tubular ^2^		Author
			ornamental		red			
	* *Salvia elegans*	Herb	Ornamental	Nectar	Lavender	Tubular ^2^		Author
	* *Salvia madrensis*	Herb	Ornamental	nectar	yellow	Tubular ^2^		Author
	* *Salvia* Indigo Spires	Herb	Ornamental	Nectar/pollen	Blue	Tubular ^2^		Author
	* *Salvia* sp. Blue	Herb	Ornamental	Nectar/pollen	Blue	Tubular ^2^		Author
	* *Salvia* sp. Pink	Herb	Ornamental	Nectar	Pink	Tubular ^2^		Author
Lythraceae								
	* *Cuphea ignea*	Herb	Ornamental	Nectar	Orange	Tubular ^2^		Author
	* *Cuphea schumannii*	Shrub	Ornamental	Nectar/pollen	Orange	Tubular ^2^		Author
	* *Cuphea* Starfire	Shrub	Ornamental	Nectar	Rose and white	Tubular ^2^		Author
	* *Cuphea* sp. Pink	Shrub	Ornamental	Nectar	Pink	Tubular ^2^		Author/ *C. ignea* X
								*angustifolia*
**Magnoliaceae**								
	* ***Magnolia*** *virginiana*	Tree	Native/	Pollen	White	Open ^3^		iNat j86
			ornamental					
Malphigiaceae								
	* *Brysonima lucida*	Shrub	Native/ ornamental	Pollen	Multi- colored	Open ^3^		Author
Malvaceae								
	***Dombeya*** *burgessiae*	Shrub	Ornamental	Pollen	Pink	Open ^3^		Author
	* *Hibiscus rosa-sinensis* X	Shrub	Ornamental	Nectar	Pink	Open ^3^		iNat whtsnnm
	* *Kosteletzkya pentacarpos*	Herb	Native	Nectar	Pink	Open ^3^		iNat ricoref ^1^
	* ***Melochia*** *tomentosa*	Shrub	Native/	Nectar	Pink	Tubular ^2^		iNat cpgibson
			ornamental					
	*Taliparitii tiliaceum*	Tree	Naturalized/	Nectar	Yellow	Open ^3^		iNat bio-joy-janae/
			ornamental					*Hibiscus tiliaceus*
Marantaceae								
	* *Thalia geniculata*	Herb	Native	Nectar	Maroon	Tubular ^2^		Author
Melastomataceae								
	* ***Centradenia*** sp.	Herb	Ornamental	Pollen	Pink	Buzz ^4^		Author
	***Dissotis*** *rotundifolia*	Herb	Ornamental	Pollen	Pink	Buzz ^4^		Author
	* ***Heterocentron*** *elegans*	Vine	Ornamental	Pollen	Pink	Buzz ^4^		Author
	***Medinilla*** *apoensis*	Shrub	Ornamental	Pollen	Pink	Buzz ^4^		Author
	* *Miconia bicolor*	Shrub	Native/	Pollen	White	Buzz ^4^		Author/
			ornamental					*Tetrazygia biolor*
	* *Tibouchina urvilleana*	Shrub	Ornamental	Pollen	Purple	Buzz ^4^		Author
	* *Tibouchina heteromalla*	Shrub	Ornamental	Pollen	Purple	Buzz ^4^		Author
**Moraceae**								
	* ***Ficus*** *altissima*	Tree	Ornamental	Stem latex F	N/A	N/A		iNat aliandbrice
Myrtaceae								
	***Melaleuca*** *quinquenervia*	Tree	Invasive/	Leaf chem M	N/A	N/A		iNat elime
			ornamental					
	* *Pimenta dioica*	Tree	Food/spice	Keaf chem M	N/A	N/A		Author
	* *Psidium guajava*	Tree	Food/fruit	Pollen	White	Open ^3^		iNat sandrae-34242
**Nelumbonaceae**								
	***Nelumbo*** *nicifera*	Herb	Ornamental	Pollen	White	Open ^3^		iNat crobinb
**Nyctaginaceae**								
	* ***Bougainvillia*** *spectabilis* X	Vine	Ornamental	Nectar	White	Tubular ^2^		Author
Oleaceae								
	***Jasminum*** *multiflorum*	Vine	Ornamental	Nectar	White	Tubular ^2^		iNat Edward PerryIV
Orchidaceae								
	* ***Prosthechea*** *radiatum*	Herb	Ornamental	Fragrance M	N/A	Gullet ^1^		Author
	***Arundina*** *graminifolia*	Herb	Ornamental	Deception	White and rose	Gullet ^1^		Author/decay on flower
	* *Cattelya mossiae*	Herb	Ornamental	Nectar	Rose	Gullet ^1^		Author
	* *Cattelya quadricolor*	Herb	Ornamental	Fragrance M	N/A	Gullet ^1^		Author
	* *Coryanthes macrantha*	Herb	Perfume orchid	Fragrance M	N/A	Unique		Author dead male in flower
	* *Coryanthes panamaensis*	Herb	Perfume orchid	Fragrance	N/A	Gullet ^1^		Author/ dead female w pollinia
								in flower
	***Dendrobium*** *anosmum*	Herb	Ornamental	Nectar	Pink and	Gullet ^1^		J. Pemberton/
					rose			Pollinia removal
	* ***Encyclia*** *oncidioides*	Herb	Ornamental	Fragrance M	N/A	N/A		Author
	* *Gongora powellii*	Herb	Perfume orchid	Fragrance M	N/A	Unique		Author
	* *Guarianthe patnii*	Herb	Ornamental	Deception	N/A	Gullet ^1^		Author
	* *Guarianthe skinneri*	Herb	Ornamental	Deception	Rose	Gullet ^1^		Author/ pollinia removal & fruit set
	* *Lycaste aromatica*	Herb	Perfume orchid	Fragrance M	N/A	Gullet ^1^		Author
	* *Lycaste cochleata*	Herb	Perfume orchid	Fragrance M	N/A	Gullet ^1^		Author/pollinia removal
	* *Mormodes rolfeana*	Herb	Perfume orchid	Fragrance M	N/A	Unique		Author/pollinia removal
	* ***Myrmecophila*** *tibicinus*	Herb	Ornamental	Fragrance M	Pink	Gullet ^1^		Author/pollinia removal
	***Phaius*** *tankervilliae*	Herb T	Ornamental	Deception	White and brown	Gullet ^1^		Author
	* *Aspasia epidendroides*	Herb	Ornamental	Fragrance M	N/A	Gullet ^1^		Author/ex adxial surface
	***Spathoglottis*** *plicata*	Herb T	Ornamental	Deception	Rose	Unique		Author
	* *Sobralia decora*	Herb T	Ornamental	Deception	Pink	Gullet ^1^		Author
	* *Sobralia sessilis*	Herb T	Ornamental	Deception	Rose	Gullet ^1^		Author
	* *Sobralia violaceae*	Herb T	Ornamental	deception	Pink	Gullet ^1^		Author
	* *Stanhopea embreei*	Herb	Perfume orchid	Fragrance M	N/A	Unique		Author
	* *Stanhopea saccata*	Herb	Perfume orchid	Fragrance M	N/A	Unique		Author
	* *Vanillla phaeantha*	Vine	Native plant	Deception	White and	Gullet ^1^		Author
					yellow			
	* *Vanillla planifolia*	Vine	Food/spice	Deception	Green and	Gullet ^1^		Thomas Trotta
					yellow			
	*Brassocattleya* Maikai	Herb	Ornamental	Nectar	Lavender and	Gullet ^1^		Author
					white			
	* *Cattleya* bifoliate var.	Herb	Ornamental	Nectar	White	Gullet ^1^		iNat Laura
								Zurro
	***Spathoglottis*** Grapette	Herb T	Ornamental	Deception	Rose	Unique		iNat Krista Clay
Passifloraceae								
	* *Passiflora prob. edulis*	Vine	Food/fruit	Nectar	White	Closed ^5^		Author
						Purple		
	* *Passiflora foetida*	Vine	Native	Nectar	White	Closed ^5^		iNat huxelyrose
	* *Passiflora* Grace Ann	Vine	Ornamental	Nectar	Red	Closed ^5^		iNat Sandra
								Demenech
	* *Passiflora* Incense	Vine	Ornamental	Nectar	Purple	Closed ^5^		Author
	* *Passiflora* Lady Margaret	Vine	Ornamental	Nectar	Red	Closed ^5^		Author
**Plumbaginaceae**								
	* ***Plumbago*** *auriculata*	Shrub	Ornamental	Nectar	Blue	Tubular ^2^		Author
	* ***Plumbago*** *scandens*	Vine	Native	Nectar	White	Tubular ^2^		Author
**Pontederaceae**								
	* ***Pontederia*** *cordata*	Herb	Native	Nectar	Blue	Tubular ^2^		Author
Rosaceae								
	* ***Rosa*** cultivar	Shrub	Ornamental	Pollen?	Rose	Open ^3^		iNat fibuzz
Rubiaceae								
	* *Arachnothryx leucophylla*	Shrub	Ornamental	Nectar/pollen	Pink	Tubular ^2^		iNat YolandaSvatik
								*Rondeletia leucophylum*
	* **Guettarda** scabra	Shrub	Native	Nectar	White	Tubular ^2^		Pimienta& Koptur 2021
	* *Hamelia longipes*	Shrub	Ornamental	Nectar/pollen	Orange	Tubular ^2^		Author
	* *Hamelia patens*	Shrub	Native/	Nectar/pollen	Orange	Tubular ^2^		Author
			ornamental					
	***Ixora*** *coccinea*	Shrub	Ornamental	Nectar/pollen	Red	Tubular ^2^		Author
	* ***Morinda*** *royoc*	Shrub	Native	Nectar/pollen	White	Tubular ^2^		Author
	***Pentas*** *lanceolata*	Herb	Ornamental	Nectar/pollen	Red, white, pink	Tubular ^2^		iNat lillybyrd
	* *Psychotria tenuifolia*	Shrub	Native	Nectar/pollen	White	Tubular ^2^		S.Lenberger specimen label
	* ***Richardia*** *grandiflora*	Herb	Weed	Nectar/pollen	Pink	Gullet ^1^		Author
Rutaceae								
	***Citrus*** sp.	Tree	Food—fruit	Nectar	White	Open ^3^		iNat kmajka
**Sapindaceae**								
	***Litchi*** *chinensis*	Tree	Food—fruit	Leaf chem M	N/A	NA		Author/brushing older leaves
**Scrophulariaceae**								
	* ***Buddleja*** *davidii*	Shrub	Ornamental	Nectar	Purple	Tubular ^2^		Author
Solanaceae								
	* *Capsicum annuum*	Herb	Food—spice	Nectar/pollen	White	Open ^3^		Author
	* *Cestrum diurnum*	Shrub	Ornamental	Nectar/pollen	White	Tubular ^2^		Author
	* *Cestrum aurantiacum*	Shrub	Ornamental	Nectar/pollen	Gold	Tubular ^2^		Author
	* *Lycianthes rantonnei*	Shrub	Ornamental	Nectar/pollen?	Purple	Open ^3^		Author
	* *Petunia* hybrida X	Herb	Ornamental	Nectar/pollen?	Purple	Gullet ^1^		Author
	* *Solanum americanum*	Herb	Native	Pollen	White	Buzz ^4^		Author
	* *Solanum bahamense*	Shrub	Native	Pollen	Blue	Buzz ^4^		Author
	* *Solanum capsicoides*	Herb	Native	Pollen	White	Buzz ^4^		Author
	* *Solanum diphyllum*	Shrub	Invasive/	Pollen	White	Buzz ^4^		Author
			ornamental					
	* *Solanum diphyllum*	Shrub	Invasive/	Leaf chem M	N/A	N/A		Author
			ornamental					
	* *Solanum donianum*	Shrub	Native	Pollen	White	Buzz ^4^		Author
	* *Solanum erianthum*	Shrub	Native	Pollen	White	Buzz ^4^		iNat Zane
								Roskoph
	* *Solanum laxum*	Vine	Ornamental	Pollen	White	Buzz ^4^		Author
	* *Solanum echinatus*	Herb	Ornamental	Pollen	Blue	Buzz ^4^		Author
	* *Solanum esculentum*	Herb	Food—vegetable	Pollen	Yellow	Buzz ^4^		author/*S. mamosum*
	* *Solanum melongena*	Herb	Food—vegetable	Pollen	Blue	Buzz ^4^		author/Grape, Roma
								Everglades
	* *Solanum pseudocapsicum*	Shrub	Ornamental	Pollen	White	Buzz ^4^		Author
	* *Solanum pyracanthos*	Herb	Ornamental	Pollen	Blue	Buzz ^4^		Author
	* *Solanum quitoense*	Herb	Food—fruit	Pollen	White	Buzz ^4^		Author
	* *Solanum seaforthianum*	Vine	Invasive/	Pollen	Blue	Buzz ^4^		Author
			ornamental					
	* *Solanum sisymbriifolium*	Herb	Naturalized/	Pollen	White	Buzz ^4^		Author
			ornamental					
	* *Solanum tampicense*	Herb	Invasive weed	Pollen	White	Buzz ^4^		Author
	* *Solanum torvum*	Shrub	Invasive weed	Pollen	White	Buzz ^4^		Author
	* *Solanum wendlandii*	Vine	Ornamental	Pollen	Blue	Buzz ^4^		Author
	* *Solanum wrightii*	Tree	Ornamental	Pollen	Lavender	Buzz ^4^		Author
	* *Solanum verbascifolium*	Shrub	Native	Pollen	White	Buzz ^4^		Author
	* *Solanum viarum*	Herb	Invasive weed	Pollen	White	Buzz ^4^		Author
**Saururaceae**								
	* ***Saururus*** *cernuus*	Herb	Native	Leaf chem M	N/A	N/A		Author
**Surianaceae**								
	* ***Suriana*** *martima*	Shrub	Native	Nectar	Yellow	Open ^3^		iNat riki-Bonema
**Turneraceae**								
	* ***Turnera*** *subulata*	Herb	Ornamental	Nectar	White	Open ^3^		Author
	* ***Turnera*** *ulmifolia*	Herb	Naturalized/	Nectar	Yellow	Open ^3^		Author
			ornamental					
Verbenaceae								
	* ***Duranta erecta***	Shrub	Ornamental	Nectar/pollen	Purple	Tubular ^2^		Author
	* ***Lantana*** *camara*	Shrub	Invasive/	Nectar/pollen	Yellow and	Tubular ^2^		Author/*D. repends*
			ornamental		lavender			
	* ***Lantana*** *involucrata*	Shrub	Native	Nectar/pollen	White	Tubular ^2^		Author
	* ***Petrea*** *volubilis*	Vine	Ornamental	Nectar/pollen	Purple	Tubular ^2^		iNat
								Sharonforsyth
	* *Stachytarpheta caatingensis*	Herb	Ornamental	Nectar/pollen	Blue	Tubular ^2^		Author
	* *Stachytarpheta cayennensis*	Herb	Ornamental	Nectar/pollen	Purple	Tubular ^2^		Author
	* *Stachytarpheta jamaicensis*	Herb	Native	Nectar/pollen	Purple	Tubular ^2^		Author
	* *Stachytarpheta mutabilis*	Herb	Ornamental	Nectar/pollen	Orange	Tubular ^2^		Author
	* *Stacytarpheta urticifolius*	Herb	Naturalized/	Nectar/pollen	Purple	Tubular ^2^		Author
			ornamental					
**Vitaceae**								
	* ***Parthenocissus*** *quinquefolia*	Vine	Native	Leaf chem M	N/A	N/A		iNat alianbrice
Zingiberaceae								
	*Alpinia zerumbet*	Herb	Ornamental	Nectar	White and	Gullet ^1^		Author
					yellow			
	*Alpinia* sp.	Herb	Ornamental	Nectar	White and	Gullet ^1^		Author
					maroon			

## Data Availability

The source data are in Table 1.

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
