# Peer review of "Plant Resource Use and Pattern of Usage by the Naturalized Orchid Bee (Euglossa dilemma: Hymenoptera: Apidae) in Florida"

_insects, 2023, doi:10.3390/insects14120909_

Round 1

Reviewer 1 Report

Comments and Suggestions for Authors

This is an important and interesting study, a report on the potential incredible impact the introduced orchid bee has on the pollination of plants in south Florida! The author has observed the invasion over the last two decades and presents here a compendium of all observations he and others have made, including presenting the bees with various potted plants to see if the bees utilize their rewards.  I think this will be of interest to many biologists interested not only in plant/animal interactions, but the importance of invaders in a changing world catches the attention of many! 

The manuscript was slightly repetitive, and the wording a bit cumbersome in parts. I have taken the liberty of commenting directly on the pdf using the comments tool in Adobe, to correct grammar and spelling mistakes, and suggest a few changes to clarify the presentation of the results. 

Comments on the Quality of English Language

clearly a native English speaker, but maybe not an English major!

Author Response

I very much appreciate the detailed, expert review which improved the clarity and grammar of the manuscript. I have followed all but a few minor suggestions. I have highlighted all of the word changes in red and have used the comment box to discuss the few places where I did not follow the suggestions of the reviewer. Please find the fully revised manuscript attached here. The revised manuscript has my comments related to both reviewers. Please see the attached file Response to reviewersInsects for my detailed response Thank you.

Reviewer 2 Report

Comments and Suggestions for Authors

Dear author,

It is with pleasure that I reviewed your work. I am impressed by this very complete work ! But I have some few suggestions to improve the understanding and the quality of the ms. It is especially related to the design of the Table 1 which is central result in the great work that you built.

It could be interesting to directly introduce the species name (Hymenoptera:Apidae) in the title, simple summary and abstract because I find difficult to understand if the study investigates on one or several Euglossa species.

L16, L17-18, L21: The italic name of the species name needs to be checked in all the ms.

L52-53: Please, add the year of the description of E. dilemma after "Bembé & Eltz"

L77: change "." by ","

L83: Same, add the year of description from Friese 

L166: "[39.40]" ==> "[39,40]"

Table 1 : Rewrite the legend of the Table 1 to be suitable to the author'instruction of Insect's journal, change the footnote position and bolding. Also, if it possible, try to extend the size of the table. The asterisk is not explained in the Table 1 legend.

L.363-364: Why this is interesting to use Asteraceae ? 

Kinds regards,

Author Response

I very much appreciate the reviewer's review which helped improved the clarity of the manuscript.  I have followed the reviewer's suggestions but have some comments related to them which, are posted as comment boxes on the manuscript itself at the place of the suggestion. The reviewer suggested that I review the author instructions for the title of the table. The author's instructions simply say that it should be a short explanatory title and caption. I looked at some examples of published papers and saw the simple explanatory titles for tables. My table needs the two descriptions of what bolding in species and family names mean and what an asterisk in front of names mean.  The fully revised manuscript is attached with word changes showing in red.

Thank you for your help! 
